# Non-canonical Wnt signalling modulates the endothelial shear stress flow sensor in vascular remodelling

Claudio A Franco[1,2]*, Martin L Jones[1], Miguel O Bernabeu[3,4],
Anne-Clemence Vion[1,5], Pedro Barbacena[2], Jieqing Fan[6], Thomas Mathivet[7,8],
Catarina G Fonseca[2], Anan Ragab[1], Terry P Yamaguchi[9], Peter V Coveney[4],
Richard A Lang[6], Holger Gerhardt[1,5,7,8,10,11]*

[1]Vascular Biology Laboratory, Lincoln's Inn Laboratories, London Research Institute, The Francis Crick Institute, London, United Kingdom; [2]Instituto de Medicina Molecular, Faculdade de Medicina Universidade de Lisboa, Lisbon, Portugal; [3]Centre for Medical Informatics, Usher Institute, The University of Edinburgh, Edinburgh, United Kingdom; [4]Centre for Computational Science, Department of Chemistry, University College London, London, United Kingdom; [5]Max-Delbrück-Center for Molecular Medicine, Berlin, Germany; [6]The Visual Systems Group, Division of Pediatric Ophthalmology, Cincinnati Children's Hospital Medical Center, Cincinnati, United States; [7]Vascular Patterning Laboratory, Vesalius Research Center, Leuven, Belgium; [8]Department of Oncology, Vascular Patterning Laboratory, Vesalius Research Center, Leuven, Belgium; [9]Cancer and Developmental Biology Laboratory, Center for Cancer Research, NCI-Frederick, National Institutes of Health, Frederick, United States; [10]German Center for Cardiovascular Research, Berlin, Germany; [11]Berlin Institute of Health, Berlin, Germany

*For correspondence: cfranco@medicina.ulisboa.pt (CAF); holger.gerhardt@mdc-berlin.de (HG)

**Competing interests:** The authors declare that no competing interests exist.

**Abstract** Endothelial cells respond to molecular and physical forces in development and vascular homeostasis. Deregulation of endothelial responses to flow-induced shear is believed to contribute to many aspects of cardiovascular diseases including atherosclerosis. However, how molecular signals and shear-mediated physical forces integrate to regulate vascular patterning is poorly understood. Here we show that endothelial non-canonical Wnt signalling regulates endothelial sensitivity to shear forces. Loss of Wnt5a/Wnt11 renders endothelial cells more sensitive to shear, resulting in axial polarization and migration against flow at lower shear levels. Integration of flow modelling and polarity analysis in entire vascular networks demonstrates that polarization against flow is achieved differentially in artery, vein, capillaries and the primitive sprouting front. Collectively our data suggest that non-canonical Wnt signalling stabilizes forming vascular networks by reducing endothelial shear sensitivity, thus keeping vessels open under low flow conditions that prevail in the primitive plexus.

## Introduction

Functional blood vessel networks are essential for vertebrate development, tissue growth and organ physiology (*Potente et al., 2011*). Vessel assembly and sprouting establish the major axial vessels and a primary network, which undergoes extensive remodelling to become functional. Also in the adult, vascular networks can be reactivated, expanded to meet changing metabolic demands, or remodelled, as a consequence of injury or local occlusion (*Carmeliet, 2005*; *Potente et al., 2011*). A

**eLife digest** Blood vessels play an essential role in growth and development as they transport many important molecules that help cells to survive. Throughout life, the forces that act on the blood vessels help to remodel the vessel network to ensure that blood gets to the parts of the body that need it. For example, the movement of blood across the surface of the endothelial cells that line the inside of the blood vessels applies a force called "shear stress" to the cells. The endothelial cells respond and adapt to the stress by altering their shape, patterns of gene activity and internal organization (known as their polarity).

It was not fully understood exactly how the forces acting on endothelial cells help to remodel the blood vessel network. Franco et al. have now investigated how a signalling pathway known as non-canonical Wnt signalling affects the remodelling of blood vessels in mice, and found that this pathway stabilizes existing connections between vessels.

Disrupting non-canonical Wnt signalling, by genetically engineering mice to lack proteins called Wnt5a and Wnt11, increased the sensitivity of endothelial cells to shear stress. Franco et al. then built a computer model that simulates blood flow and endothelial cell polarity in a network of blood vessels; this enabled them to measure the endothelial cells' response to blood flow in complex vascular networks. The model was then used to show that endothelial cells lacking non-canonical Wnt signalling are able to reorient and become polarized against the direction of blood flow at lower levels of shear stress. Thus, non-canonical Wnt signalling helps to raise the threshold of shear stress above which endothelial cells change their properties.

Further work is now needed to identify how non-canonical Wnt signalling interferes with the ability of the endothelial cells to sense shear stress levels.

large number of mouse mutants have been described as having defects in vascular remodelling. Yet, in contrast to vascular sprouting, very little is known about the intrinsic cellular and molecular mechanisms controlling vascular remodelling. One aspect of remodelling is vessel segment regression, in which existing connections are lost. Endothelial cell (EC) death drives programmed regression of the ocular hyaloid vessels (*Lobov et al., 2005*) and pupillary membrane (*Meeson et al., 1999*). Whilst a similar mechanism was thought to be driving developmental vascular remodelling, recent reports proposed that vessel segment regression in the remodelling retinal blood vessels involves dynamic rearrangement of ECs, which actively migrate from regressing vessel segments to integrate into neighbouring vessels (*Franco et al., 2015*; *Udan et al., 2013*). Chen et al. postulated that ECs in zebrafish brain vessels sense a threshold of low blood flow below which vessel regression is triggered irreversibly (*Chen et al., 2012*). Our recent rheology modelling of the retinal plexus also predict regression of poorly perfused vessel segments (*Bernabeu et al., 2014*; *Franco et al., 2015*), and demonstrated that EC axial polarization against the blood flow direction is a conserved feature in remodelling vessels (*Franco et al., 2015*). Our observations lead us to propose that flow-induced EC polarization directs migration of ECs that reside in low flow or oscillatory flow segments towards juxtapose high flow segments. As a consequence, this movement of ECs between vessel segments with differential flow regimes leads to regression of low-flow branches and stabilization of the higher flow segments (*Franco et al., 2015*). Blood flow is critical for vascular remodelling (*Hahn and Schwartz, 2009*), but the relevance and the mechanistic understanding of how physical forces and signalling pathways collectively stabilize or disrupt vessel connections remains unknown.

Here we show that ECs use non-canonical Wnt ligands in a short-range, paracrine manner to stabilize connections during vascular remodelling. We show that loss of endothelial-derived Wnt5a and Wnt11 sensitizes ECs to polarize against the blood flow direction at lower levels of wall shear stress, in vitro and in vivo, thereby leading to premature and excessive vessel regression in mouse. We postulate that the enhanced sensitivity to flow in non-canonical Wnt-deficient endothelium promotes earlier discrimination of flow asymmetries between neighbouring vessel segments in the capillary plexus, thus driving premature vessel regression and accelerated remodelling.

# Results

## Endothelial-derived Wnt ligands protect from premature vessel regression independent of apoptosis

Wnt/β-catenin signalling has been shown to both promote and inhibit vessel regression (*Lobov et al., 2005*; *Phng et al., 2009*). To gain further insight into the role of Wnt ligands in vessel regression, we conditionally inactivated Wnt-ligand secretion by recombination of the floxed *Wls/Evi/Gpr177* allele (*Carpenter et al., 2010*) in ECs (Pdgfb-iCreERT2 or Tie2-Cre). *Wls* encodes for a transporter chaperone protein required for secretion of all Wnt ligands (*Banziger et al., 2006*; *Bartscherer et al., 2006*). Embryonic endothelial-specific *Wls* deletion (*Wls*fl/fl::Tie2-Cre) leads to mid-gestation lethality, demonstrating an important vascular function for endothelial-derived Wnt ligands (*Table 1*). Tamoxifen-inducible *Wls* deletion in ECs (Pdgfb-iCreERT2::*Wls*fl/fl, hereafter *Wls* iEC-KO) led to significantly decreased vascular density compared to littermate controls (*Figure 1a*). Quantification revealed increased regression profiles (quantified by the Col.IV-sleeves and Icam2-breakage profiles), while sprout frequency, proliferation, EC density and apoptosis rates were unaffected (*Figure 1b* and *Figure 1—figure supplement 1a,b*). Surprisingly, these results did not recapitulate recent findings by Korn et al. who reported that *Wls* iEC-KO causes increased vessel regression through increased apoptosis (*Korn et al., 2014*). Our recent work established that regression in the mouse vasculature follows a sequence of events that begin with vessel stenosis, followed by cell retraction that finally leads to resolution, leaving only empty matrix behind (*Franco et al., 2015*). The frequency distribution of regression profiles at these distinct stages of segment regression, i.e. stenosis, retraction or resolution, was similar in *Wls* iEC-KO and *Wls* WT mice (*Figure 1c*) indicating that the lack of secretion of Wnt ligands from ECs affects the frequency but not the mechanism of vessel regression. Experimental hyperoxia-induced vessel obliteration (which is driven by endothelial apoptosis (*Alon et al., 1995*) caused similar central capillary network dropout in *Wls* WT and *Wls* iEC-KO, suggesting that EC Wnt-ligands are not able to significantly protect from endothelial cell apoptosis-mediated vessel regression events (*Figure 2a,b*). Defects in pericyte recruitment have been linked to increased vessel instability and vessel regression (*Benjamin et al., 1998*). We analysed pericyte coverage using NG2 marker and observed no significant changes between *Wls* WT and *Wls* iEC-KO retinas (*Figure 3a,b*).

## Endothelial non-canonical Wnt ligands prevent premature vessel regression

RT-PCR profiling on RNA extracts from isolated P7 retinal ECs (*Figure 4a*) identified expression of Wnt ligands associated with canonical (*Wnt3*, *Wnt3a*, *Wnt6*, *Wnt7b*, *Wnt9a* and *Wnt10a*) and non-canonical (*Wnt5a* and *Wnt11*) Wnt signalling. Expression of the canonical Wnt/β-catenin-dependent targets *Axin2*, *CyclinD1* and *Lef1* (*Clevers and Nusse, 2012*) were unaffected in *Wls* iEC-KO (*Figure 4b,c*), and nuclear Lef1 levels were even slightly increased (*Figure 4d*). Intercrossing the canonical Wnt signalling reporter mouse BAT-gal (*Maretto et al., 2003*) also revealed no differences in X-gal positive ECs (*Figure 4e*). Also expression of endothelial Dll4/Notch signalling components, potentially influenced by canonical Wnt signalling (*Corada et al., 2010*), was unaffected (*Figure 4b, c*). Together, these findings identify that canonical Wnt signalling is intact in *Wls* iEC-KO, and

**Table 1.** Tie2-Cre *Wls*fl/fl embryos die at mid-gestation. Table showing number of embryos collected at the specified embryonic time point post-coitum (E) and at birth. Relative frequency each genotype of embryos/pups is shown as percentage.

♂ **Tie2-Cre::*Wls* fl/wt X ♀ *Wls* fl/fl**

| Genotype | E12.5 | % | E13.5 | % | E16.5 | % | Adults | % |
|---|---|---|---|---|---|---|---|---|
| *Wls*fl/fl | 6 | 0.25 | 4 | 0.29 | 7 | 0.41 | 20 | 0.38 |
| *Wls*fl/wt | 5 | 0.21 | 4 | 0.29 | 4 | 0.24 | 13 | 0.25 |
| *Wls*fl/fl::Cre+ | 6 | 0.25 | 4 | 0.29 | 4 (3 dead) | 0.24 | 0 | 0.00 |
| *Wls*fl/wt::Cre+ | 7 | 0.29 | 2 | 0.14 | 2 | 0.12 | 20 | 0.38 |

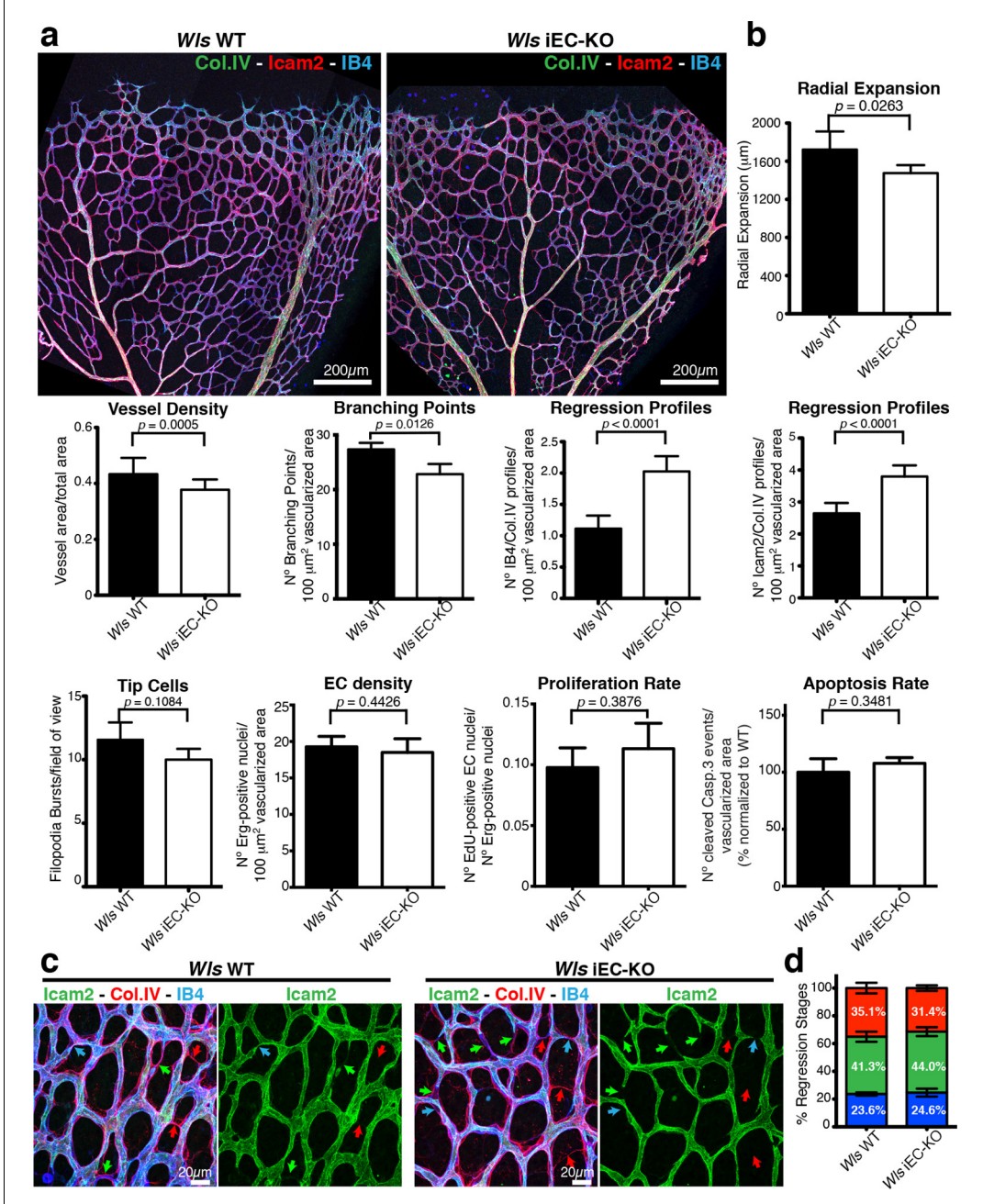

**Figure 1.** EC-derived Wnt ligands protect against vessel regression. (a) Overview of retinal vascular plexus of P6 control and *Wls* iEC-KO mice, labeled for lumen (ICAM2), ECs (IB4) and ECM (Col.IV). (b) Quantification of vascular parameters demonstrating that increased vessel regression is a main feature of *Wls* iEC-KO in P6 retinas. IB4/Col.IV regression profiles correspond to the number Col.IV-positive segments negative for IB4 staining. ICAM2/Col.IV regression profiles correspond to number of Col.IV-positive vessel segments partially or totally negative for ICAM2 staining. p values from unpaired, two-tailed t-test. Mean +/-SEM; N = 6 mice; 3 litters. (c) High magnification images of the vascular plexus of control and *Wls* iEC-KO mice marked for each stage of vessel regression (stenosis, blue arrows; retraction, green arrow; resolution, red arrows). (d) Quantification of the each specific vessel regression stage in *Wls* iEC-KO and *Wls* WT P6 retinas (stenosis, blue bars; retraction, green bars; resolution, red bars). Two-way ANOVA with Sidak multiple comparisons test. Mean +/-SEM; N = 5 mice; 3 litters; N = 288 and N = 398 regression events for *Wls* WT and *Wls* iEC-KO mice, respectively.

The following figure supplement is available for figure 1:

**Figure supplement 1.** *Wls* iEC-KO mice show normal proliferation and apoptosis rates.

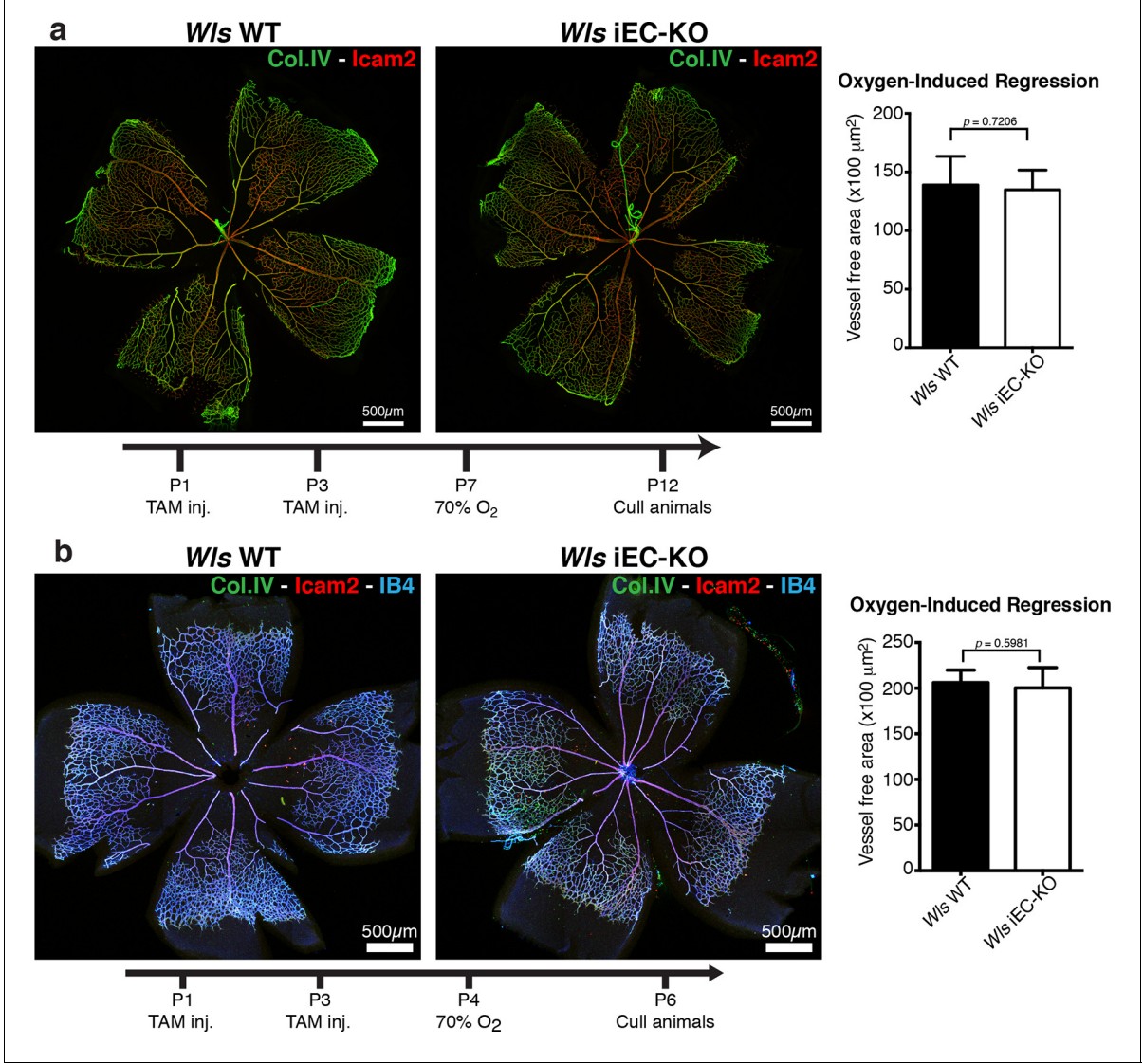

**Figure 2.** Oxygen-induced vessel regression is not enhanced in *Wls* iEC-KO mice. (**a**) Representative confocal images of *Wls* iEC-KO and *Wls* WT P12 mouse retinas under 70% oxygen concentration from P7 until P12, and labeled for vessel lumen (Icam2, red) and extracellular matrix (Col.IV, green). Quantification shows no significant difference in area of vessel obliteration between *Wls* iEC-KO and *Wls* WT mouse. p values from unpaired, two-tailed t-test. Mean +/-SEM; N = 6 mice; 2 litters. (**b**) Representative confocal images of *Wls* iEC-KO and *Wls* WT P16 mouse retinas under 70% oxygen concentration from P4 until P6, and labeled with vessel lumen (Icam2, red), endothelial cells (IB4, blue) and extracellular matrix (Col.IV, green). Quantification shows no significant difference in area of vessel obliteration between *Wls* iEC-KO and *Wls* WT mouse. p values from unpaired, two-tailed t-test. Mean +/-SEM; N = 6 mice; 3 litters.

suggest that the observed increase in regression was possibly due to loss of endothelial non-canonical Wnt signalling.

Indeed, endothelial-specific *Wnt5a* inactivation (*Wnt5a*^fl/fl^::Pdgfb-iCreERT2, hereafter *Wnt5a* iEC-KO) led to increased vessel regression, decreased vascular density and a mild decrease in radial vascular expansion (*Figure 5a,b*). Constitutive *Wnt11* KO mice showed a milder phenotype with a slight decrease in radial expansion, but no significant differences in vascular density (*Figure 5a,b*). However, compound *Wnt5a* endothelial-specific KO and *Wnt11* KO mice, named *Wnt5a* iEC-KO; *Wnt11* KO hereafter, largely phenocopied the vascular defects of *Wls* iEC-KO mice (*Figure 6a,b*). As in *Wls* iEC-KO mice, ECs numbers and apoptosis rate were unaffected in *Wnt5a* iEC-KO; *Wnt11* KO (*Figure 6a,b*). Also the tracheal vasculature, undergoing post-natal remodelling (*Baffert et al., 2004*), showed a significant decrease in vascular density in *Wnt5a* iEC-KO; *Wnt11* KO, and an

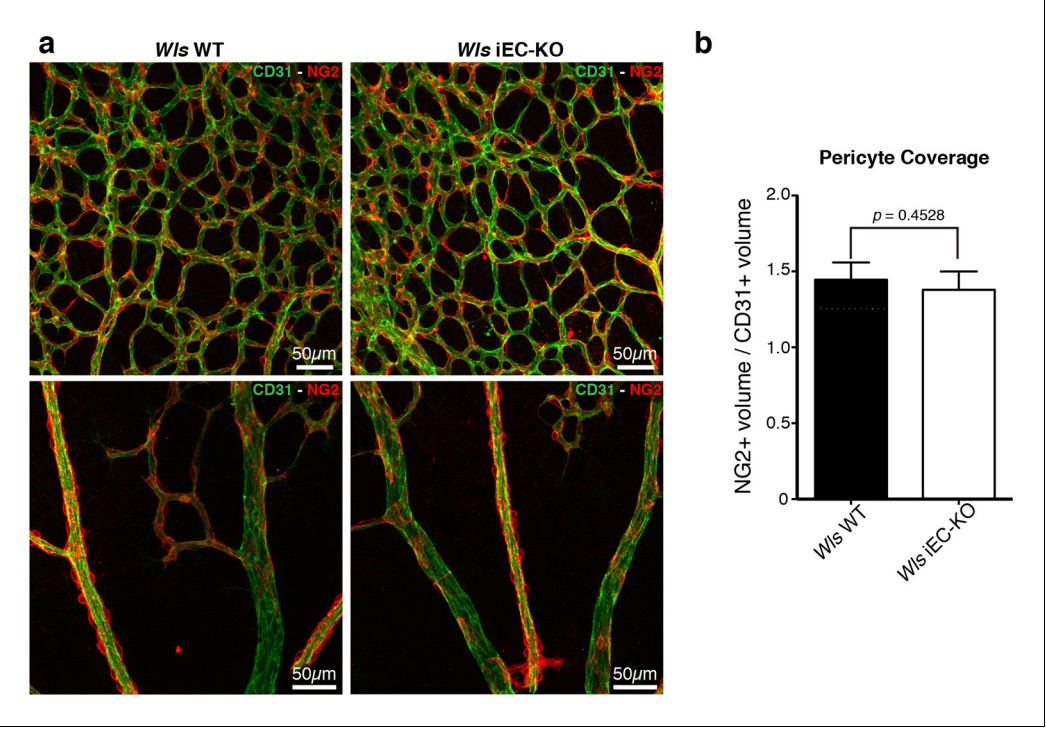

**Figure 3.** Normal pericytic coverage in *Wls* iEC-KO. (**a**) Overview of retinal vascular plexus of P6 control and *Wls* iEC-KO mice, labeled for endothelial cells (CD31), and pericytes (Ng2), in capillary plexus (top panels) and main vessels (bottom panels). (**b**) Quantification of pericyte coverage of retinal blood vessels, showing no significant change between control and *Wls* iEC-KO in P6 retinas. p values from unpaired, two-tailed t-test. Mean +/-SEM; N = 4 mice; 2 litters.

associated increase in vessel regression events (*Figure 6c,d*). We conclude that endothelial-derived non-canonical Wnt ligands prevent excessive and premature vessel disconnection.

## Non-canonical Wnt signalling modulates endothelial response to flow-dependent wall shear stress

We recently showed that EC nucleus-to-Golgi axial polarity predicts migration patterns at sites of vessel regression in vivo, and that differential flow/shear patterns in juxtaposed vessels drive asymmetries in cellular movements, thus promoting stabilization of high-flow and regression of low-flow vessel segments (*Franco et al., 2015*). Further, we have developed a computational approach to simulate blood flow in retinal networks and calculate wall shear stress and flow patterns (*Bernabeu et al., 2014*). Given the known involvement of non-canonical Wnt signalling in Planar Cell Polarity (PCP) and cell polarization (*Devenport, 2014*; *Segalen and Bellaiche, 2009*) we hypothesized that *Wls* iEC-KO could have defects in cell polarization. We therefore analysed if non-canonical Wnt signalling could influence coordinated polarization of ECs in vivo and in response to flow. We stained *Wls* iEC-KO and control mice for Golgi, lumen, and EC nuclei and extracted maps of axial polarity for entire retinal vascular networks (*Franco et al., 2015*), in a novel analysis methodology that we call hereafter Polarity Network (PolNet) analysis (*Figure 7a* and *Figure 7—figure supplement 1*). To measure efficiency of endothelial polarization in response to flow, we calculated the angle between the axial polarity vectors and the predicted flow vectors (*Figure 7b*). Similar to controls, *Wls* iEC-KO EC cells significantly polarize against blood flow direction across all assessed regions of the network (*Figure 7c*). Surprisingly, *Wls* iEC-KO ECs in capillaries, the regions of active remodelling, showed significantly better polarization against the blood flow compared to control cells (*Figure 7d*). When plotting the percentage of cells polarized within 45 degrees of anti-parallel orientation to flow against the computationally predicted wall shear stress, WT and *Wls* iEC-KO cells segregated such that Wnt-deficient cells reached 60 percent of cells polarizing already at less than

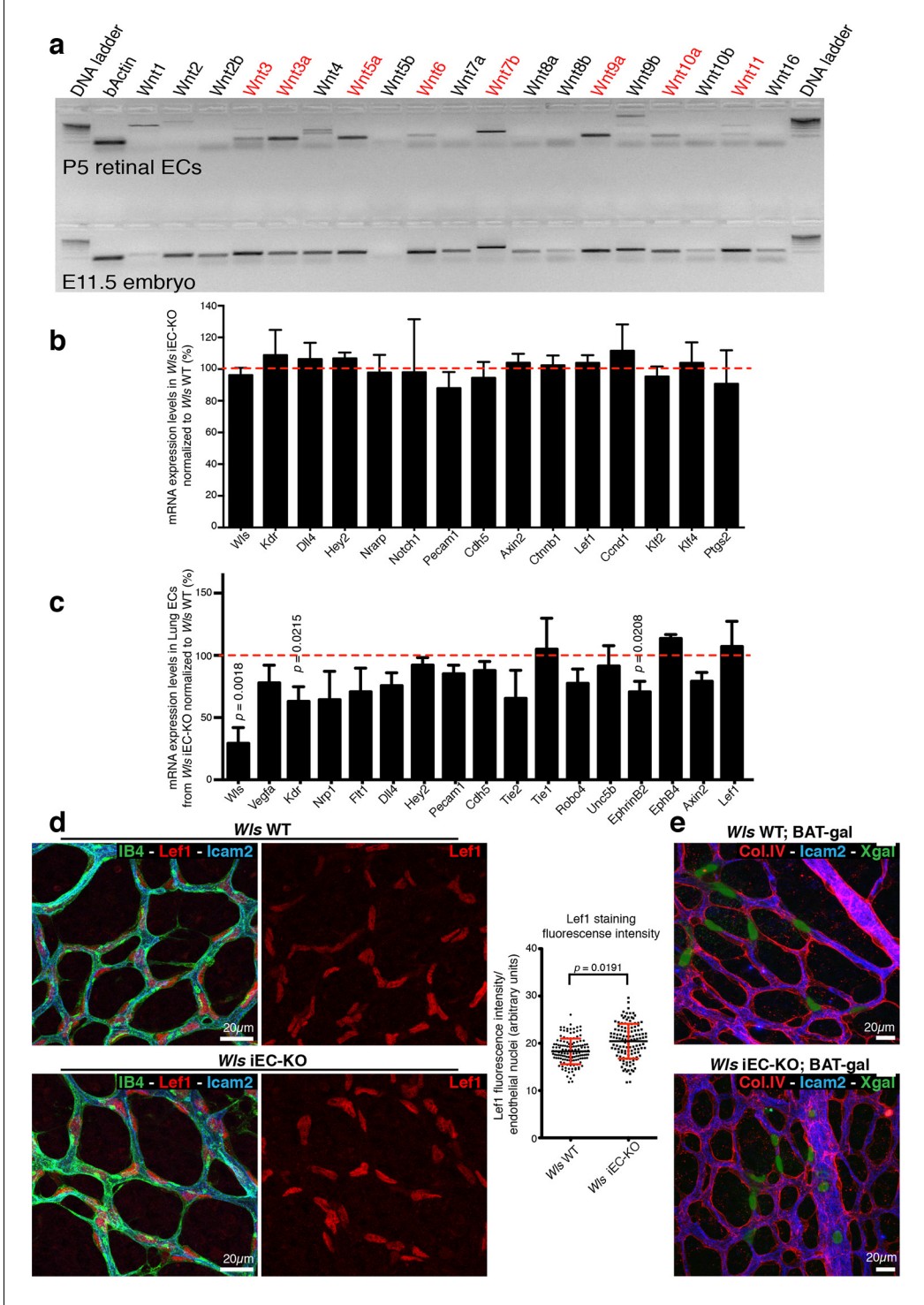

**Figure 4.** *Wls* iEC-KO show no significant defects in canonical Wnt signalling. (a) RT-PCR for all known mouse Wnt ligands using mRNA extracts from isolated retinal endothelial cells of P7 wild-type retinas (red text represents positive bands). (b) Semi-quantitative real-time analysis of mRNA expression levels of different genes in P6 *Wls* iEC-KO retinas normalized to *Wls* WT retinas from whole retina extracts. p values from unpaired, two-tailed t-test. Mean +/- SD; N = 4 mice; 2 litters. (c) Semi-quantitative real-time analysis of mRNA expression levels of different genes in P6 *Wls* iEC-KO normalized to *Wls* WT from isolated lung endothelial cells. p values from unpaired, two-tailed t-test. Mean +/-SEM; N = 3 mice; 2 litters. (d) Lef1 immunostaining and quantification of fluorescence intensity (graph right) in endothelial cells (IB4) from *Wls* iEC-KO and *Wls* WT retinas. p values from unpaired, two-tailed t-test. Mean +/- SEM; N = 234 *Wls* WT cells and N = 128 *Wls* iEC-KO cells; 4 mice. (e) X-gal staining, indicative of canonical Wnt signalling activation, in
*Figure 4 continued on next page*

*Figure 4 continued*

*Wls* iEC-KO; BAT-gal and *Wls* WT; BAT-gal retinas. No correlation was found with X-gal positive cells and regression profiles (visualized by ECM (Col.IV) and lumen (Icam2) stainings).

4 Pa whereas WT cells needed more than 7 Pa of shear to reach 60 percent (*Figure 7e*). These data suggest that loss of endothelial-derived non-canonical Wnt ligands substantially increases the sensitivity of cells to shear.

To directly investigate whether non-canonical Wnt ligands regulate the sensitivity of ECs to polarize against the flow, we used a microfluidic device to test endothelial polarization in cultured monolayers exposed to laminar flow induced shear stress (*Ziegler and Nerem, 1994*). Intriguingly, siRNA-mediate knockdown of Wnt5a and Wnt11 in HUVECs led to a significant increase in polarization against the blood flow in this reductionist in vitro system suggesting that loss of endothelial Wnts directly affects aspects of endothelial cell biology involved in shear sensing and/or transduction (*Figure 8a,b*).

In addition to polarizing cells, flow triggers a well-documented transcriptional response of a number of genes including *Klf2, Klf4*, and *Ptgs2* (*Hahn and Schwartz, 2009*). Moreover, VE-cadherin, VEGFR2 and PECAM1 have been described to act in a complex sensing and relaying flow-mediated shear forces (*Tzima et al., 2005*). To understand whether Wnt affected a more general sensitivity to flow, we therefore studied transcriptional levels following Wnt5a and Wnt11 knockdown. Interestingly, HUVECs depleted of Wnt5a and Wnt11 reacted transcriptionally to flow in the same order of magnitude as control cells for all genes analysed (*Figure 8c*). These data indicate that the mechanism of sensing and transducing flow signals into axial polarization differs from the mechanism triggering the transcriptional responses, such that endothelial Wnt-ligands only affect the former but not the latter.

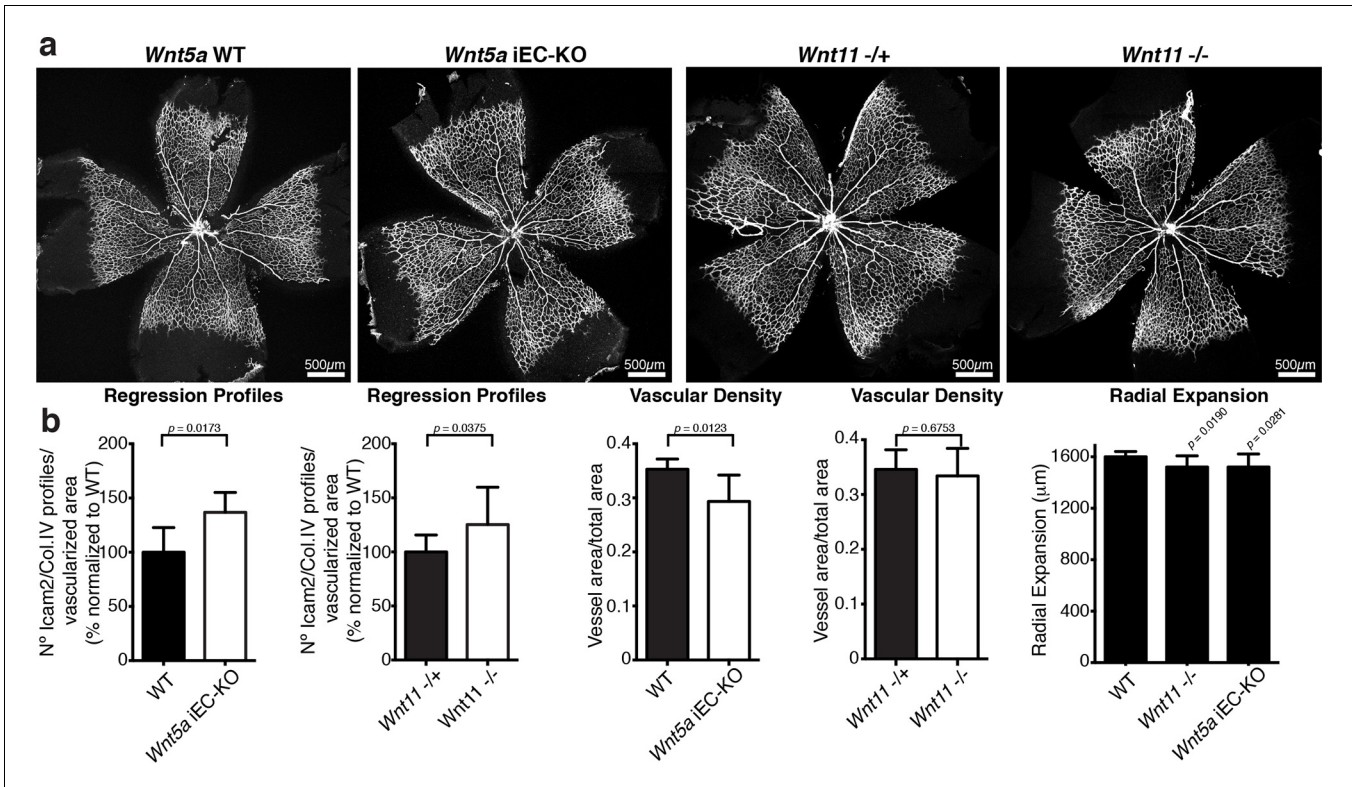

**Figure 5.** Characterization of vascular parameters in *Wnt5a* iEC-KO and *Wnt11* KO retinas. (**a**) Overview of retinal vascular plexus of P6 *Wnt5a* iEC-KO, *Wnt11* KO, and corresponding control mice, labeled with ECM (Col.IV, grey). (**b**) Quantification of different vascular parameters in P6 retinas on the different mouse strains. p values from unpaired, two-tailed t-test. Mean +/-SEM; N = 5 mice; 3 litters.

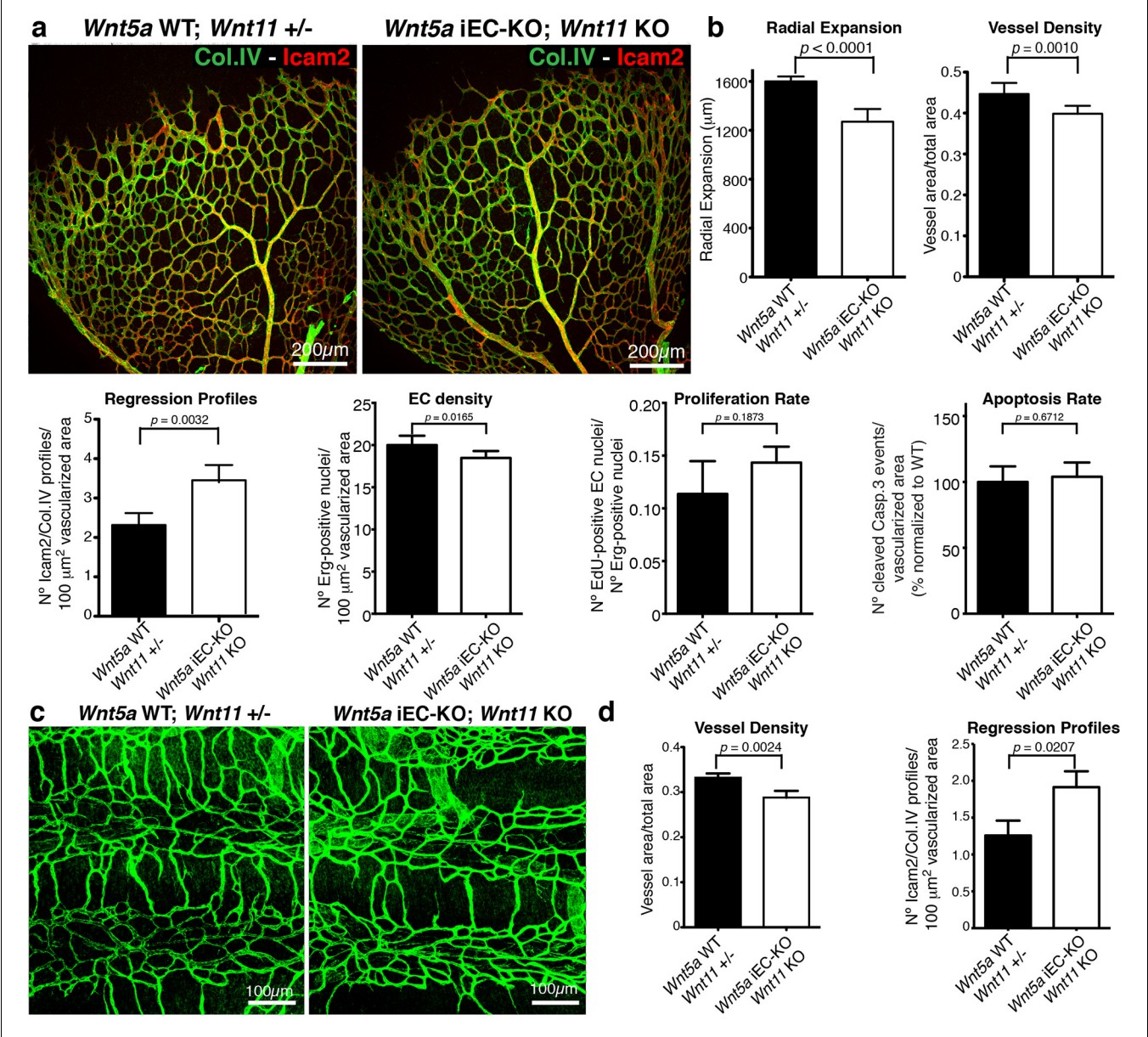

**Figure 6.** Non-canonical Wnt signalling regulates vessel regression. (**a**) Overview of retinal vascular plexus of control and compound *Wnt5a* iEC-KO; *Wnt11* KO mice, labelled with lumen (Icam2) and ECM (Col.IV) markers. (**b**) Quantification of different vascular parameters showing increased vessel regression in *Wnt5a* iEC-KO; *Wnt11* KO retinas. p values from unpaired, two-tailed t-test. Mean +/-SEM; N = 7 mice; 3 litters. (**c**) Overview of trachea vascular plexus of P6 *Wnt5a* iEC-KO; *Wnt11* KO and control mice labelled for CD31 (green). (**d**) Quantification of vessel density and regression profiles in the trachea of *Wnt5a* iEC-KO; *Wnt 11* KO and WT P6 mice. p values from unpaired, two-tailed t-test. Mean +/-SEM; N = 4 mice; 2 litters.

To understand how increasing shear influences the remodelling process in dependence on Wnt signalling, we injected systemically angiotensin II to increase blood flow and therefore augment wall shear stress levels. Quantification of regression profiles showed an increase in the number of regression profiles in retinas from both *Wls* WT and *Wls* iEC-KO mice (***Figure 9a,b***), with angiotensin II-treated control mice having similar numbers of regression profiles as non-treated *Wls* iEC-KO mice, with a corresponding decrease in vessel density (***Figure 9b***). We then used our previously described PolNet analysis to evaluate polarisation patterns of ECs after systemic angiotensin II treatment (***Figure 9c***). Notably, we observed a significant increase in the polarisation of endothelial cells against the blood flow direction in the capillary network (***Figure 9d***). Thus, increasing shear forces,

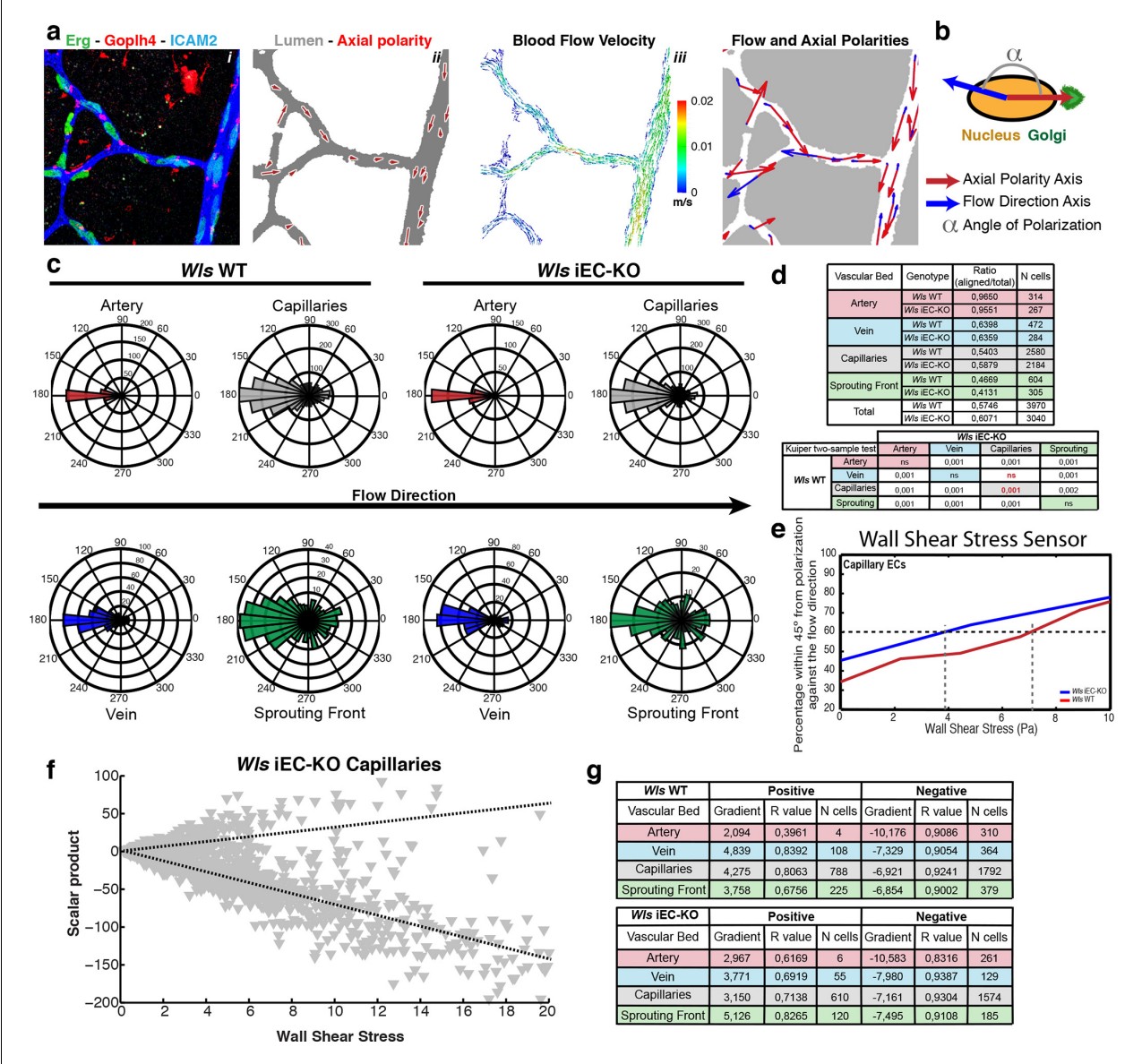

**Figure 7.** Endothelial-derived Wnt ligands modulate endothelial polarization in response to wall shear stress. (a) Example of axial polarity of an *Wls* iEC-KO P6 retina, labeled for EC nuclei (Erg), lumen (Icam2) and Golgi (Golph4) (i), corresponding image segmentation of the vascular plexus with axial polarity vectors in red (ii), flow pattern simulation of the selected area (iii), and correlation between axial polarity and blood flow direction at the endothelial nuclear position (iv). (b) Representation of the principle of angle calculation between axial and flow polarities of endothelial nuclei in (a), highlighting the lumen of blood vessels (grey), and the axial polarity of all ECs (red arrows). (c) Analysis of the endothelial axial polarity angle in the main vessels, relative to predicted blood flow direction by the rheology in silico model in *Wls* WT and *Wls* iEC-KO mice (n = 3 retinas). (d) Quantitative analysis of the percentage of ECs polarized at 180°(± 45°) degrees compared to the flow direction in the different vascular beds of *Wls* WT and *Wls* iEC-KO mice (n = 3 retinas). (e) Correlative analysis of wall shear stress and EC polarization in the capillary vascular bed of *Wls* WT and *Wls* iEC-KO mice. (f) Representative graph showing the distribution of scalar products in function to wall shear stress levels for ECs from *Wls* iEC-KO capillaries. Scalar product corresponds to the product between length of the axial polarity vector and the cosine of the angle between the axial polarity vector and the flow direction vector. (g) Linear regression analysis of positive (polarized with flow) and negative (polarized against the flow) scalar product points for each endothelial cell nucleus. Gradient, R-value and number of cells analyzed for each vascular bed and genotype are shown. N = 3 retinas.

The following figure supplement is available for figure 7:

**Figure supplement 1.** Endothelial polarization patterns in *Wls* iEC-KO.

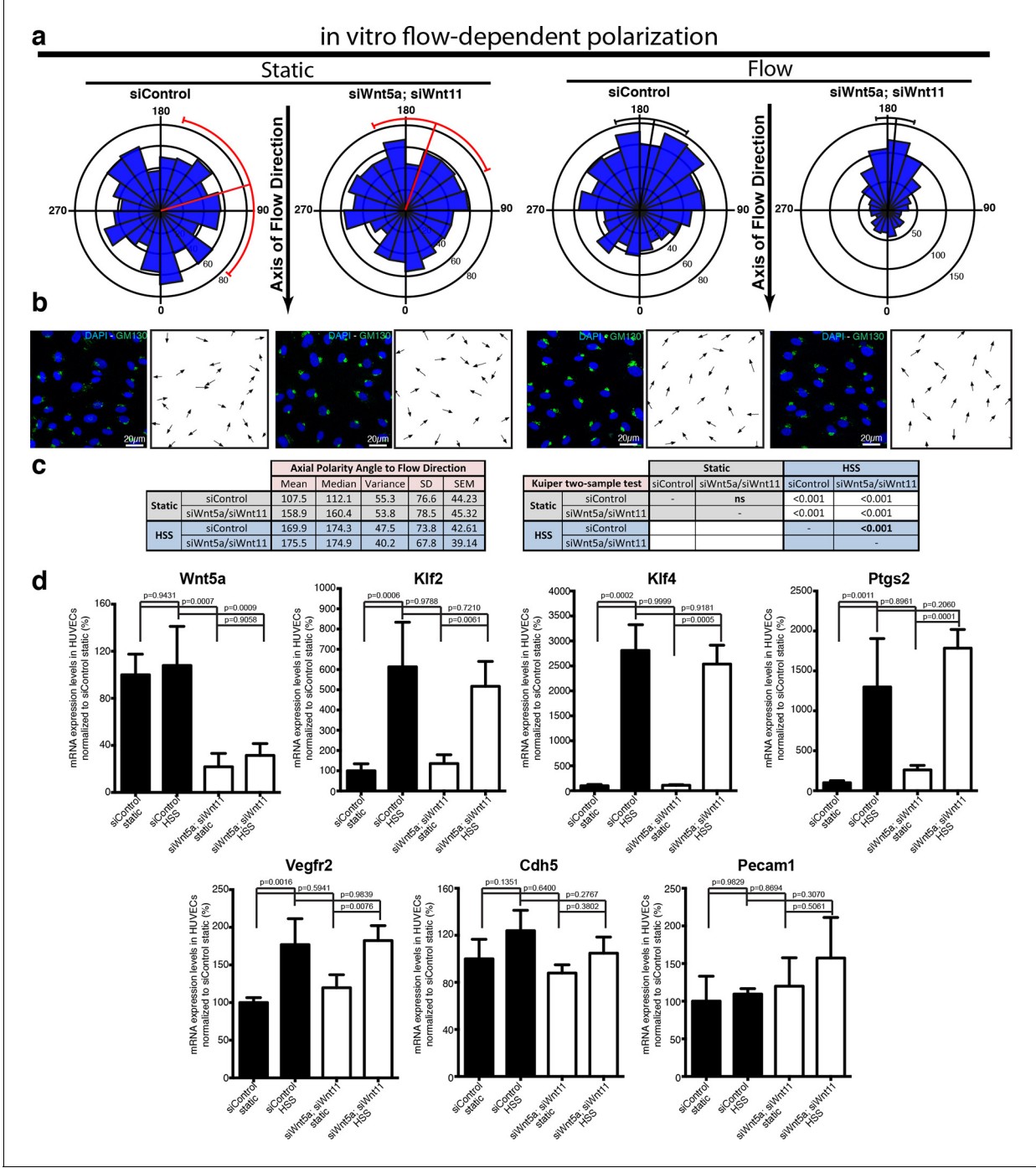

**Figure 8.** Non-canonical Wnt signalling modulates flow-induced polarity but not flow-induced transcriptional gene expression response. (a) Rose-plot representation of axial polarity of ECs treated with Control or *Wnt5a* and *Wnt11* specific siRNAs in static conditions or in response to 2 Pa flow in a microfluidic device (n = 3 independent experiments). Line running from the center represents the Mean of the dataset. Arcs extending to either side represent the 95% confidence limits of the mean (red means not significantly polarised; black means significantly polarised). (b) Representative images of endothelial cell polarity in flow chamber stained for nuclei (Dapi, Blue) and Golgi apparatus (GM130, green), and corresponding axial polarity vectors (black arrows). (c) Quantitative analysis of EC polarization related to the flow direction in the microfluidic device. p values from non-parametric two-tailed Kuiper's test. (d) Semi-quantitative real-time analysis of mRNA expression levels of different genes in Control or *Wnt5a* and *Wnt11* specific siRNAs in static or stimulated with 2 Pa conditions in a microfluidic device. p values from one-way ANOVA with multiple comparisons. Mean +/-SD; N = 3 independent experiments.

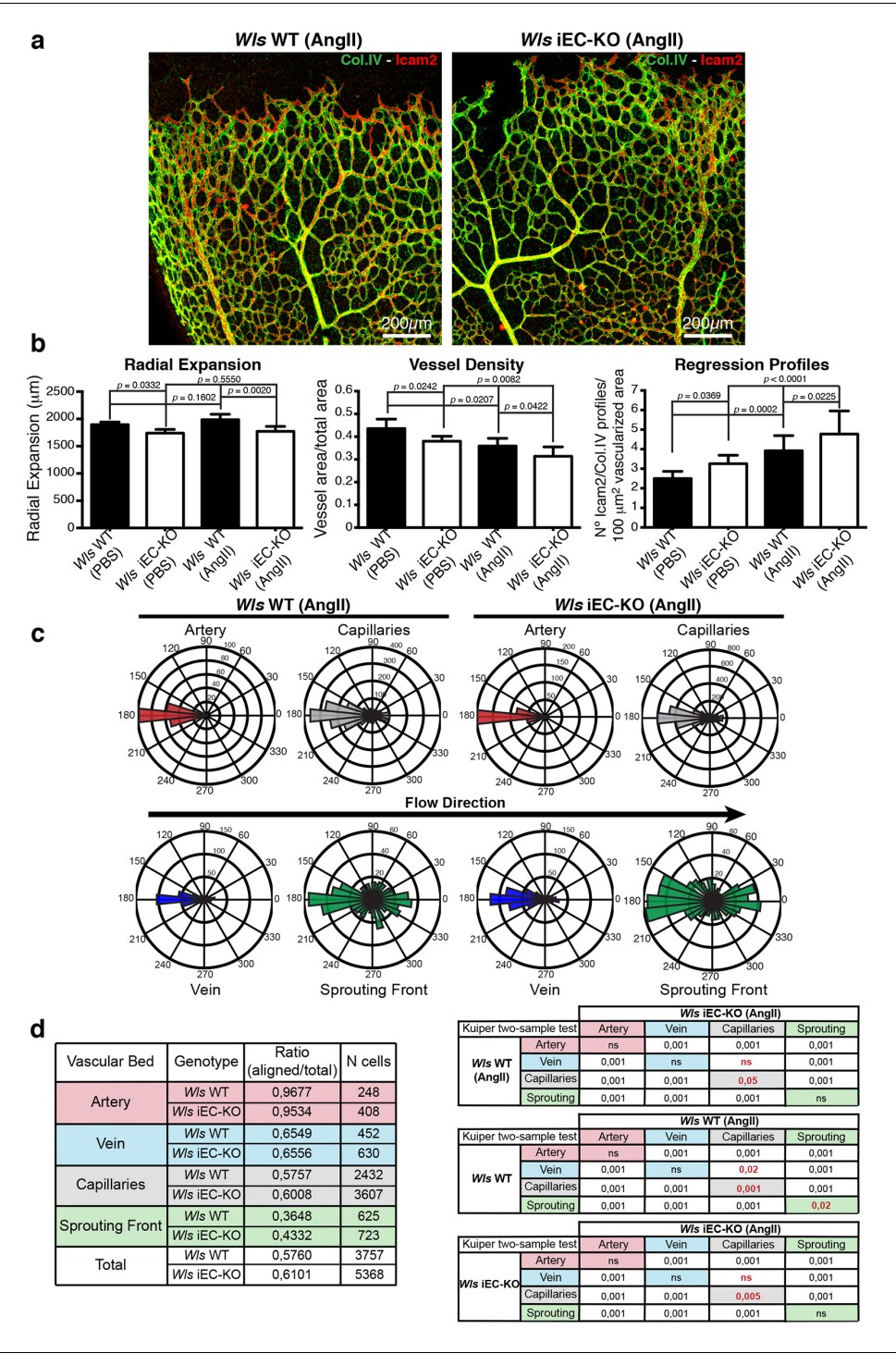

**Figure 9.** Systemic Angiotensin-II treatment accelerates vessel regression independent of Wnt signalling. (**a**) Overview of retinal vascular plexus of P7 control and *Wls* iEC-KO mice treated with angiotensin II, labelled for lumen (Icam2) and ECM (Col.IV). (**b**) Quantification of radial expansion, vessel density and regression profiles in angiotensin II-treated and PBS-treated control retinas. p values from one-way ANOVA with Holm-Sidak's multiple comparison test. Mean +/-SEM; N = 4 mice; 3 litters. (**c**) Analysis of the endothelial axial polarity angle in the main vessels, correlated to predicted blood flow direction by the rheology in silico model in *Wls* WT and *Wls* iEC-KO mice treated with angiotensin II. (**d**) Quantitative analysis of the percentage of ECs polarized at 180°(± 45°) degrees compared to the flow direction in the different vascular beds of *Wls* WT and *Wls* iEC-KO mice with and without angiotensin II treatment (n = 3 retinas).

or the sensitivity of the endothelium to shear-induced polarization appears to have the same effect on remodelling, and can act synergistically.

Finally, we asked whether increasing non-canonical Wnt signalling can lower the sensitivity to shear, and thus prevent vessel regression. Surprisingly, inducible endothelial overexpression of *Wnt5a* (hereafter *Wnt5a* OE) had no effect on vessel regression and vascular density under normal conditions (*Figure 10a–c*), illustrating that increased and sustained *Wnt5a* levels are not sufficient to prevent physiological vessel remodelling. Moreover, *Wnt5a* OE failed to prevent the angiotensin-II-

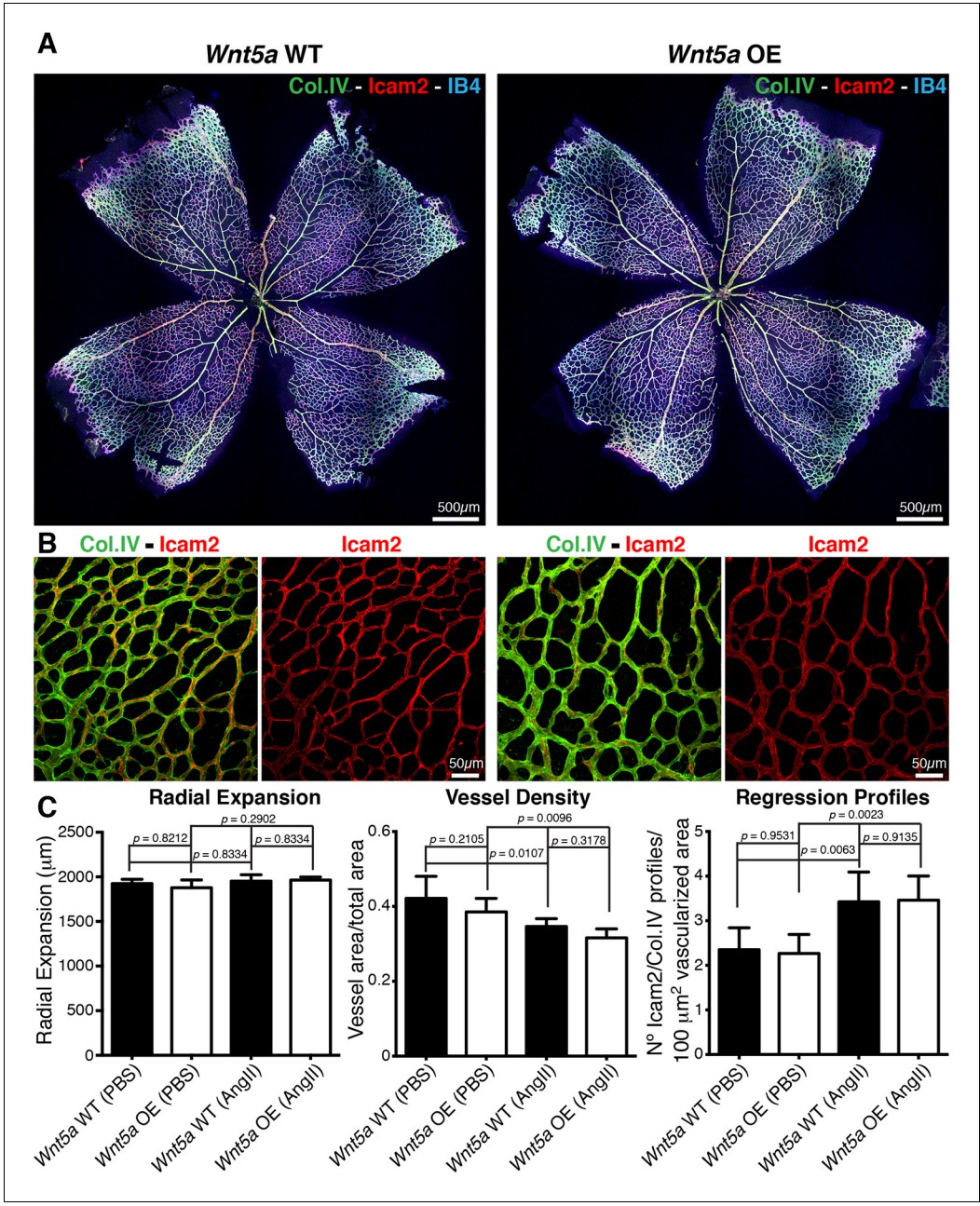

**Figure 10.** Overexpression of *Wnt5a* in endothelial cells does not inhibit vessel regression. (a) Overview of retinal vascular plexus of P6 endothelial-specific *Wnt5a* GOF and corresponding control mice, labeled for ECM (Col.IV, green), blood vessels (IB4, blue) and vascular lumen (Icam2, red). (b) Higher magnification of retinal vascular plexus of P6 endothelial-specific *Wnt5a* GOF and corresponding control mice, labeled for ECM (Col.IV, green), blood vessels (IB4, blue) and vascular lumen (Icam2, red). (c) Quantification of vascular parameters showing no significant differences between *Wnt5a* GOF and corresponding control retinas, with indicated treatments. p values from one-way ANOVA with Holm-Sidak's multiple comparison test. Mean +/-SEM; N = 4 mice; 2 litters.

mediated increase in vessel regression and decrease in vessel density (*Figure 10a–c*). Thus, our results show that non-canonical Wnt signalling is not used as a mechanism to drive flow-dependent vessel regression, but its absence sensitizes ECs to flow-induced regression.

## Discussion

### Endothelial cell-autonomous control of developmental vessel regression onset

Our present results show that ECs have intrinsic molecular mechanisms that regulate the flow-dependent shear stress response in vivo, which are important to control cell polarity and vessel regression in vascular remodelling. We show that ECs secrete non-canonical Wnt ligands (Wnt5a and Wnt11) that decrease the capacity of ECs to orientate against the direction of blood flow specifically in vascular capillaries.

Interestingly, we found that non-canonical Wnt signalling acts as a permissive rather than an instructive cue. Deleting *Wnt5a* and *Wnt11* did not change the overall pattern of the vascular network, and overexpressing *Wnt5a* in ECs of mouse retinas did not affect normal levels of vessel regression or vascular morphology. Thus, our results suggest that flow-induced vessel regression governs the overall mechanism that selects which vessel segments are redundant and non-functional to undergo regression, and that a basal level of non-canonical Wnt signalling is present to decrease the sensitivity of ECs to the flow-dependent remodelling program (*Figure 11*).

Our results go against a recent proposal by Korn et al., who concluded that non-canonical Wnt signalling was involved in the regulation of ECs survival and apoptosis (*Korn et al., 2014*). We find no evidence for decreased EC proliferation, or increased EC apoptosis upon loss of *Wls* or *Wnt5a/ Wnt11*, in both *in vivo* and *in vitro* experiments. Korn et al. used TNP-470 to inhibit non-canonical Wnt signalling, a compound which inhibits the broad-spectrum enzyme methionine aminopeptidase-2, and that has been suggested to also interfere with VEGF signalling, a major regulator of cell proliferation and survival (*Emoto et al., 2000*; *Sin et al., 1997*). Our study instead used selective genetic loss-of-function, potentially explaining some of the differences. Our results point to a distinct

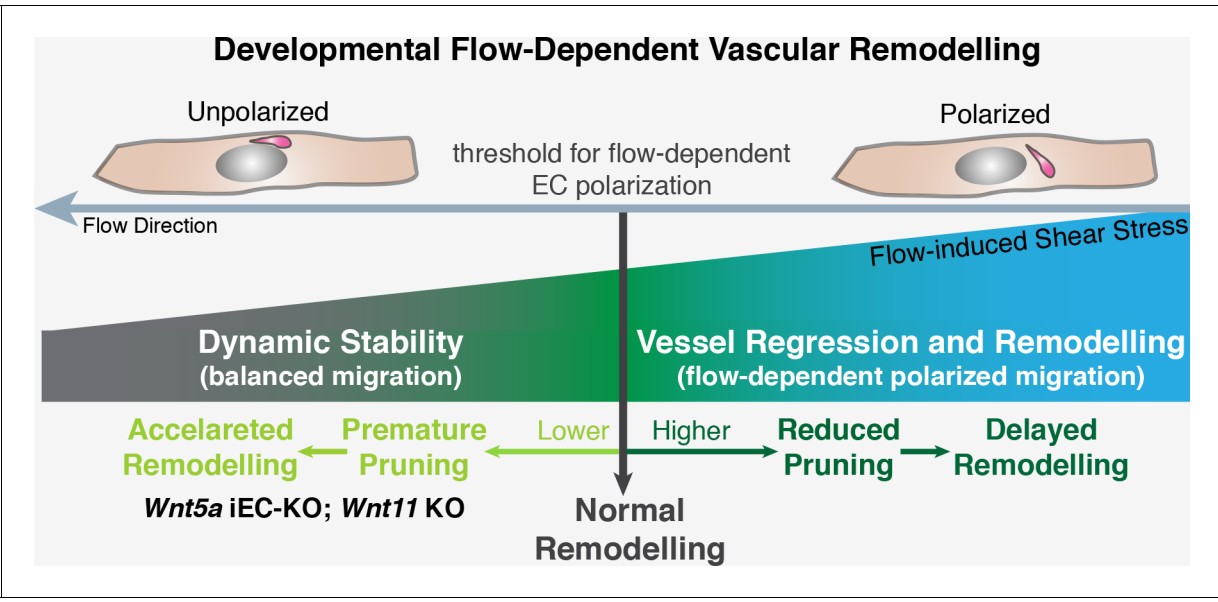

**Figure 11.** Working model for non-canonical Wnt signalling regulation of flow-induced remodeling. Initiation of the vascular remodeling program is dependent on the level of wall shear stress (threshold) to which ECs robustly polarize against the flow direction. Below the threshold, ECs movements in the immature plexus are balanced by countermovements of adjacent ECs maintaining vessel connections, in a state of dynamic stability. When exposed to shear stress levels above the response threshold, ECs react by polarizing and migrating against the flow direction, triggering vessel regression and vessel remodeling. Signaling pathways influencing the threshold for flow-dependent EC polarization can delay remodeling (raising the threshold) or induce premature remodeling (lowering the threshold), as in the case of deficient non-canonical Wnt signaling.

effect of non-canonical Wnt signalling on EC polarity in response to flow. In the light of recent reports showing that developmental vessel regression is driven by cell migration/rearrangements under the influence of flow (*Chen et al., 2012*; *Franco et al., 2015*; *Kochhan et al., 2013*; *le Noble et al., 2004*; *Lenard et al., 2015*; *Sato et al., 2010*; *Udan et al., 2013*), our observed effects on EC polarity are likely the main driving force for enhanced vessel regression in our mouse mutants.

## Non-canonical Wnt signalling controls vessel regression by modulating the threshold for flow-induced EC polarization

The levels of vessel regression in both wild-type and *Wls* iEC-KO mice could be manipulated by administration of angiotensin II, a potent vasoconstrictor. Interestingly, angiotensin II-treated wild-type animals showed similar levels of vessel regression as untreated *Wls* iEC-KO mice, and the same drug could further raise regression levels in *Wls* iEC-KO mice. This data further suggests that non-canonical Wnt signalling effects in vessel regression are indeed blood flow-dependent. But most importantly, it suggests the existence of a shear stress threshold for the onset of vessel regression. Our quantitative analysis combining EC axial polarity patterns and *in silico* flow simulations shows that with increasing shear more cells are polarized better, further advocating for the presence of a shear threshold. We propose that non-canonical Wnt signalling regulates this threshold. How mechanistically Wnt5a and Wnt11 controls this threshold remains unresolved. Interestingly, Martin Schwartz group showed that HUVECs have a defined threshold to polarize parallel to flow, and that VEGFR3 acts as a modulator of blood flow shear response by regulating the flow sensor VEGFR2/VEcadherin/Pecam1 (*Baeyens et al., 2015*). Non-canonical Wnt signalling could interact with this pathway and thus impact on the flow shear stress sensor. However, our analysis demonstrated that Wnt5a and Wnt11-depleted HUVECs activate expression of key components in flow sensing (*Klf2*, *Klf4*, *Ptgs2*) in the same order of magnitude as control cells, suggesting that non-canonical Wnt signalling modulates the physical reorganization of cell polarity rather than flow sensing itself.

## Implications for vascular remodelling

The profound motility and rearrangement of ECs in the immature vascular plexus (*Chen et al., 2012*; *Franco et al., 2015*; *Jakobsson et al., 2010*; *Sato et al., 2010*) implies that ECs need to coordinate their cellular movements in order to maintain vessel integrity and vessel connections. We propose that the primitive network before flow onset, or at a subthreshold level of flow, is in a state of dynamic stability where the movement of cells is less directional, but balanced by countermovements of adjacent cells such that the network remains open and lumenised. In this context, non-canonical Wnt signalling is likely to facilitate coordinated EC behaviour balancing cell movements in low-flow segments. Flow-induced polarity will supersede this mechanism driving ECs to polarize against the flow direction. It is tempting to speculate that flow-independent cell rearrangements and flow-induced cell movements stand in some form of competition to each other. Whereas the balancing rearrangements act to maintain vessels open even under low-flow conditions, flow-induced polarization introduces a bias in the system that leads to stenosis and regression of low-flow vessel segments. Thus, one could assume that forces or signals that drive cells to maintain the vessel open need to be overcome by the flow induced polarization event. What drives the rearrangements of cells in the primitive plexus and how flow in one segment initiates regression in another is poorly understood.

Recent results show that VE-cadherin organizes the junctional and cortical actin cytoskeleton, (*Sauteur et al., 2014*), and that differential VE-cadherin dynamics drive cell rearrangements (*Bentley et al., 2014*). Cells with higher VEGF signalling and lower Notch activity show increased mobility by displaying a larger mobile fraction of VE-cadherin at their junctions (*Bentley et al., 2014*). Whether this also holds true for events during regression is unclear. However, given that Notch is also active in remodelling (*Ehling et al., 2013*; *Lobov et al., 2011*), VE-cadherin is a component of EC-to-EC and fluid shear stress force sensing (*Conway et al., 2013*), and that VE-cadherin is implicated in coordinating endothelial polarity in collective migration (*Vitorino and Meyer, 2008*), it is tempting to speculate that rearrangements in the primitive plexus involve Notch signalling as a driver of differences in cell motility, and that non-canonical Wnt works as a signalling pathway to balance net movements through coordination of cell cohesion, enabling symmetry of movements. Flow will break this symmetry in the primitive network as it

provides an extrinsic directional signal that will polarize EC movements preferentially out of the low-flow segments and into the high-flow segments.

## Materials and methods

### Mice and treatments

The following mouse strains were used: $Wls^{fl}$ (*Carpenter et al., 2010*); Pdgfb-iCreERT2 (*Claxton et al., 2008*); $Wnt5a^{fl}$ (*Miyoshi et al., 2012*); $Wnt11^{-}$ (*Majumdar et al., 2003*); Bat-Gal (*Maretto et al., 2003*); R26mTmG (*Muzumdar et al., 2007*); and Wnt5a GOF (unpublished, provided by T. Yamaguchi, details will be published in a different report). Mice were maintained at the London Research Institute under standard husbandry conditions. Tamoxifen (Sigma, Germany) was injected intraperitoneally (IP) (20 μl/g of 1 mg/mL solution) at postnatal day 2 (P2) before eyes were collected at P5 onwards. In mosaic recombination experiments tamoxifen was injected (20 μl/g of 0.04 mg/mL solution) at P3 before eyes were collected at P6, as described previously (*Franco et al., 2013*).

For EC proliferation assessment in the retina, mouse pups were injected IP 4 hr before collection of eyes with 20 ul/g of EdU solution (0.5 mg/mL; Thermo Fischer Scientific, Waltham, Massachusetts, USA, C10340). Oxygen-dependent vessel obliteration was achieved using two different regimes of hyperoxia. At P4 (regime 1) or P7 (regime 2) pups were place in 70% oxygen chamber until P6 (regime 1) or P12 (regime 2). Animals were sacrificed immediately after hyperoxia treatment and processed for retinal vasculature analysis. Angiotensin II (10 mg/mL in PBS; Sigma) was injected IP 10 μL/g daily at P4, P5 and P6 and pups were culled at P7. Control mice were injected using PBS alone. Animal procedures were performed in accordance with the Home Office Animal Act 1986 under the authority of project license PPL 80/2391. The investigators were not blinded to allocation during experiments and outcome assessment and the experiments were not randomized.

### Cell culture and microfluidic chamber experiments

HUVECs (PromoCell, Germany) were routinely cultured in EGM2-Bulletkit (Lonza, Switzerland) and mycoplasma tested. For siRNA experiments, HUVECs were transfected with ON-TARGET smart pool control untargeting (D-001210-02-20) and siRNAs against human *Wnt5a* (L-003939-00-0005), *Wnt11* (L-009474-00-0005), were purchased from Dharmacon (Lafayette, Colorado, USA). HUVECs were transfected with 25 nM siRNA using the Dharmafect 1 transfection reagent following Dharmacon protocols. Twenty-four hours post transfection HUVECs were plated at confluence in IBIDI slides (height 0.6 mm, IBIDI, Germany). Sixteen hours later, unidirectional laminar shear stress (SS) was applied using peristaltic pumps (Gilson, France) connected to a glass reservoir (ELLIPSE, France) and to the IBIDI slide. Local shear stress was calculated using Poiseuille's law and averaged 2 Pa. Cells were exposed to shear stress for 4 hr using EGM2 media (Lonza) and then fixed using 100% cold-methanol for 10 min and washed three times in PBS. Cells were then stained for Golgi (GM130, 1/500, BDBiosciences, Franklin Lakes, NJ) and nucleus (DAPI, 1/10000, Sigma). Polarity of cell was evaluated looking at the angle formed by the nucleus-Golgi main axe compared to flow direction. Orientation of the cell was evaluated by looking at the angle of major axe of the nucleus compared to flow direction. For each experiment, five fields containing more than 90 cells have been analysed.

### Immunofluorescence

Eyes were collected from P5 onwards and fixed with 2% PFA in PBS for 5 hr at 4°C, thereafter retinas were dissected in PBS. Blocking/permeabilisation was performed using Claudio's Blocking Buffer (CBB) (*Franco et al., 2013*), consisting of 1% FBS (Thermo Fisher Scientific), 3% BSA (Sigma), 0.5% triton X100 (Sigma), 0.01% Na deoxycholate (Sigma), 0,02% Na Azide (Sigma) in PBS pH = 7.4 for 2–4 hr at 4°C on a rocking platform. Primary and secondary antibodies were incubated at the desired concentration in 1:1 CBB:PBS at 4°C overnight in a rocking platform. A list of primary and corresponding secondary antibodies can be found in *Supplementary file 1*. Dapi (Sigma) was used for nuclei labeling. Retinas were mounted on slides using Vectashield mounting medium (Vector Labs, H-1000, Burlingame, CA). For imaging we used a Carl Zeiss LSM780 scanning confocal microscope (Zeiss, Germany).

## FACS of endothelial cells from neonatal mouse retinas

Retinas from neonatal P7 mice were dissected in cold PBS and rinsed in PBS. Retinal cells were dissociated with 1 mg/ml Collagenase A (Roche, Germany, 10103578001), 3 U/ml DnaseI (Roche) in DMEM (Thermo Fisher Scientific) at 37°C for 30 min and cell suspension was passed through cell strainer. After several washes in PBS, cell suspension was incubated with PE rat anti-mouse CD31 (BD Biosciences, 553373) and APC rat Anti-Mouse CD45 (BD Biosciences,561018) antibodies on ice for 15 min. Cells were then washed and applied to FACS. RNA from CD31+CD45- cells was extracted using QIAGEN microRNA kit (Netherlands). PCR for the different Wnt ligands were performed with standard PCR protocols using primers listed in *Supplementary file 2*.

## Dynabead-mediated isolation of lung endothelial cells from neonatal mice

To isolate murine endothelial cells we adapted the protocol described in Sun et al. (*Sun et al., 2012*). Briefly, lungs of P6 mouse pups were removed, minced using forceps, and digested with 1% Collagenase-A supplemented with DNAseI (3 U/mL) in HBSS with calcium/magnesium for 60 min at 37°C. Digested tissues were passed through a 14-gauge needle and then filtrated through a 40-μm cell strainer. The cell suspension was incubated for 45 min at 4°C with 50 μl of magnetic Dynabeads (Thermo Fisher Scientific) that had been conjugated overnight with anti–mouse CD31 antibody in PBS/EDTA 2 mM and 2%BSA in a rocking platform. Cells with beads attached were collected using an MPC magnet (Thermo Fisher Scientific) and washed 6 to 8 times with PBS/EDTA 2 mM and 2% BSA. Endothelial cell fraction and non-endothelial fraction was then centrifuged and ressuspended in lysis buffer from RNeasy MiniKit (Qiagen) and stored at -80°C prior RNA isolation using RNeasy MicroKit (Qiagen).

## Gene expression assays

For mouse retina gene expression profiling, eyes were collected and retinas dissected in RNAlater (Qiagen). Retinal RNA extraction was done using RNeasy MicroKit (Qiagen). For HUVEC gene expression profiling, treated or control cells were collected directly in RLT lysis buffer from the RNeasy MicroKit (Qiagen) and further processed for RNA isolation. Reverse transcription of mRNA was performed using First-Strand cDNA Synthesis Kit (Roche) using the manufacturer recommended protocol. Semi-quantitative real time-PCR was performed using a 7900HT Fast Real-Time PCR System and Taqman gene expression probes (Applied Biosystems, Thermo Fisher Scientific). A list of primers used for gene expression profiles can be found in *Supplementary file 3*.

## Whole-mount X-Gal staining

Mouse pups eyes at the desired stage were collected in 1% PFA and kept at 4°C for 4 hr. Retinas were dissected and washed twice in PBS. X-gal staining was developed in 2 mM MgCl2 (Sigma), 0.01% Na deoxycholate (Sigma) and 0.02% Nonidet P-40 (Sigma), 5 mM K3Fe(CN)6 (Sigma), 5 mM K4Fe(CN)6 (Sigma), and 0.5 mg/mL X-gal (Promega) in PBS pH = 7.4 at 35°C in a rocking platform. After X-gal staining, retinas were processed as for further antibody stainings. X-gal signal was obtained by exciting X-gal precipitate with the helium–neon laser 633 nm wavelength.

## PolNet analysis

Given the complexity and technical aspects of the PolNet Analysis the full details related to the methodology will be published in a separate manuscript. We present here a brief description of our methodology.

First, the plexus was manually segmented using Adobe Photoshop (San Jose, CA), producing a binary mask which was subsequently skeletonized using a Voronoi diagram based method (http://uk.mathworks.com/matlabcentral/fileexchange/27543-skeletonization-using-voronoi). The local vessel diameters were calculated using maximum inscribed circles at multiple positions along each vessel segment and this information was used to construct a 3D model of the plexus (*Bernabeu et al., 2014*), code available at https://github.com/UCL/BernabeuInterface2014). The surface defined by this model was used as the input to a Lattice Boltzmann Computational Fluid Dynamics solver, HemeLB (https://github.com/UCL/hemelb), run on a High Performance Computing cluster. The raw fluorescence images were processed in MATLAB using the built-in 'Ginput' function to add points

corresponding to the nucleus and Golgi of each cell and recording their locations. These positions defined a vector with magnitude and angle describing the spatial relationship between the points. Pairs of points were recorded for each cell, one at the center of the nucleus and one at the center of the Golgi, defining a vector with a magnitude and angle describing the spatial relationship between the points. The WSS values from the flow simulation were recorded at the positions of the cell nuclei, with the WSS at each point described by a vector giving the magnitude and angle of the applied shear stress. Each plexus was subdivided into artery, vein, capillary and sprouting front regions and each cell assigned to one of these vascular beds. The angular distributions were compared using the Kuiper test (the circular statistics equivalent of the Kolmogorov Smirnov test) with each comparison yielding a p-value indicating the likelihood that the two samples are drawn from the same underlying angular distribution. The calculation was performed using the Circular Statistics Toolbox from MAT-LAB's FileExchange (*Berens, 2009*). In addition we binned the angular data according to WSS magnitude to plot the proportion of cells within 45° of being anti-aligned with the flow as a function of WSS. We calculated the scalar product of the two vectors, given by magnitude(cell polarity)*magnitude(WSS)*cos(theta) which combines information about the length and relative angles of the vectors. By plotting the scalar product versus WSS, we were able to extract a gradient corresponding to magnitude(cell)*cos(theta), i.e. the projection of the cell polarity vector onto the axis defined by the WSS vector. A larger negative gradient corresponds to a larger polarization effect for a given WSS.

## Quantification measurements and statistical analysis

Complete high-resolution three-dimensional (3D) rendering of whole mount retinas were acquired using a LSM780 laser-scanning microscope (Zeiss). Tiled scans of whole retinas were analyzed with Imaris (Bitplane, Andor Technology, United Kingdom) or ImageJ. Radial expansion corresponds to the mean distance from the optic nerve to the edge of the sprouting blood vessels (4 measurements per retina were done and averaged). Vessel density corresponds to the vascular area (measured by thresholding isolectin B4 signal in ImageJ) divided by the total area of vascularized tissue (3–5 20x objective images of regions between artery and vein were used per retina). Number of branching points was measured by manually quantifying all branching points in 3–5 20x objective images, of regions between artery and vein per retina, and dividing by the total area of vascularized tissue. Regression profiles were manually measured in 3–5 20x objective images and divided by the total area of vascularized tissue. IB4/Col.IV regression profiles correspond to the number of empty base-ment membrane collagen sleeves, i.e. Col.IV-positive segments negative for IB4 staining. ICAM2/Col.IV regression profiles correspond to number of Col.IV-positive vessel segments and segments negative for ICAM2 staining or presenting a breakage in the continuity of the luminal staining. Sprouting activity corresponds to the number of filopodia bursts, clusters of filopodia emanating from the leading edge, per field of view in 4–6 20x objective images of the sprouting front for each retina. Proliferation of endothelial cells was measure by quantifying the total number of endothelial cell nuclei (labeled by Erg immunostaining) positive for EdU staining in 3–5 20x objective images, in regions containing the sprouting front, and dividing by the total area of vascularized tissue. Quantification of apoptosis in regression profiles was measured as the number of regression profiles positive for cleaved caspase-3 and divided by the total number of regression profiles in regions used for quantification, and given as percentage.

All statistical analysis was performed using Prism 5.0 (GraphPad), Oriana 4 (Kovach Computing Services) and Matlab (Mathworks) software.

## Acknowledgements

The authors thank Dr. E Dejana, and Dr. AP McMahon for providing mouse strains. This work was supported by Cancer Research UK, the Lister Institute of Preventive Medicine, a Leducq Transatlantic Network Grant (Artemis), the EMBO Young Investigator Program, an ERC starting grant REshape 311719, the BIRAX Regenerative Medicine Initiative, the EPSRC grants2020 Science (EP/I017909/1) and Large Scale Lattice Boltzmann for Biocolloidal Systems (EP/I034602/1), UK Consortium on Meso-scale Engineering Sciences (UKCOMES) (EP/L00030X/1), and the EC-FP7 projects CRESTA (grant no. 287703) and MAPPER (http://www.mapper-project.eu/, grant no. 261507). CAF was supported by a Marie Curie Actions Fellowship of the FP7 People Program (ID:255150), FCT investigator (IF/00412/

2012) and FCT grant (EXPL/BEX-BCM/2258/2013). The funders had no role in study design, data collection and interpretation, or the decision to submit the work for publication.

## Additional information

### Funding

| Funder | Grant reference number | Author |
|---|---|---|
| Cancer Research UK | | Claudio A Franco<br>Anne-Clemence Vion<br>Anan Ragab<br>Holger Gerhardt |
| Fundacao para a Ciencia e Tecnologia | EXPL/BEX-BCM/2258/2013 | Claudio A Franco<br>Catarina G Fonseca |
| Fundação para a Ciência e a Tecnologia | LISBOA-01-0145-FEDER-007391 | Claudio A Franco |
| European Commission - Marie Curie Actions | 255150 | Claudio A Franco |
| Fundação para a Ciência e a Tecnologia | IF/00412/2012 | Claudio A Franco |
| UKCOMES | EP/L00030X/1 | Miguel O Bernabeu<br>Peter V Coveney |
| European Commission - FP7 Projects | 287703 | Miguel O Bernabeu<br>Peter V Coveney |
| European Commission - FP7 Projects | 261507 | Miguel O Bernabeu<br>Peter V Coveney |
| Engineering and Physical Sciences Research Council | EP/I017909/1 | Miguel O Bernabeu<br>Peter V Coveney |
| Engineering and Physical Sciences Research Council | EP/I034602/1 | Miguel O Bernabeu<br>Peter V Coveney |
| Lister Institute of Preventive Medicine | | Holger Gerhardt |
| Leducq Transatlantic Network | Artemis | Holger Gerhardt |
| EMBO Young Investigator Program | | Holger Gerhardt |
| European Research Council | REshape 311719 | Holger Gerhardt |
| BIRAX - Regenerative Medicine Initiative | | Holger Gerhardt |

The funders had no role in study design, data collection and interpretation, or the decision to submit the work for publication.

### Author contributions

CAF, HG, Conception and design, Acquisition of data, Analysis and interpretation of data, Drafting or revising the article; MLJ, MOB, Conception and design, Acquisition of data, Analysis and interpretation of data; A-CV, Acquisition of data, Analysis and interpretation of data, Drafting or revising the article; PB, JF, TM, CGF, AR, Acquisition of data, Analysis and interpretation of data; TPY, PVC, Analysis and interpretation of data, Contributed unpublished essential data or reagents; RAL, Conception and design, Analysis and interpretation of data, Contributed unpublished essential data or reagents

### Author ORCIDs

Claudio A Franco, http://orcid.org/0000-0002-2861-3883
Martin L Jones, http://orcid.org/0000-0003-0994-5652
Holger Gerhardt, http://orcid.org/0000-0002-3030-0384

### Ethics

Animal experimentation: Animal procedures were performed in accordance with the United Kingdom Home Office Animal Act 1986 under the authority of project license PPL 80/2391.

## Additional files

### Supplementary files

• Supplementary file 1. List of antibodies used in the immunochemistry and immunofluorescence studies

• Supplementary file 2. List PCR primers used in Wnt ligand gene expression profiling of isolated retinal endothelial cells

• Supplementary file 3. List qPCR Taqman primers used in gene expression experiments.

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
