## [Decision Letter]

Thank you for submitting your work entitled "Non-canonical Wnt signalling modulates the endothelial shear stress flow sensor in vascular remodelling" for peer review at *eLife*. Your submission has been favorably evaluated by Janet Rossant (Senior editor), and three reviewers, one of whom, Ewa Paluch, is a member of our Board of Reviewing Editors.

The reviewers have discussed the reviews with one another and the Reviewing editor has drafted this decision to help you prepare a revised submission.

Summary:

This manuscript describes the role of non-canonical Wnt signalling in developing vascular beds using the retinal as a model system. The authors propose and test the hypothesis that non-canonical Wnt signaling reduces EC sensitivity to shear stress and thereby modulates vessel regression in forming vascular networks, stabilizing vessels under low flow conditions.

Essential revisions:

The reviewers find the paper interesting and solid, however several important points need to be addressed.

1) While the interpretation of the data proposed by the authors could makes sense, the authors should explore alternative explanations of their observations by examining a possible effect of perivascular cells. Indeed, it could be that reduction of non-canonical Wnt signaling alters support from adjacent perivascular cells, which affects the response of endothelial cells to reduced flow. Along the same lines, angiotensin II might affect pericyte migration and it is not clear that its effect can be attributed exclusively to changes in blood flow (have the authors actually measured blood flows upon addition of angiotensin?)

2) Specificity of the cre lines used: the authors should provide evidence to support the fact the knockouts are confined to ECs

3) Could the authors better link the two systems they use and somehow connect the observations in vivo and in the HUVEC cells? Do the changes in transcription in HUVECs actually match to any degree what is seen in vivo? Alternatively, could more physiological conditions be applied to the HUVEC assay? 20 dyn/cm2 is much higher than what is typically observed in vivo, would the changes in gene expression observed also occur for lower shear stresses? Could the authors also provide some images of the HUVEC system so the reader gets a better idea of what is meant by polarity in Figure 7?

4) In Figure 9 the authors report that overexpressing Wnt5a does not prevent vessel regression. How about in the presence of angiotensin? Would overexpression of Wnt5a rescue the effect of increased blood flow?

---

## [Author Response]

*1) While the interpretation of the data proposed by the authors could makes sense, the authors should explore alternative explanations of their observations by examining a possible effect of perivascular cells. Indeed, it could be that reduction of non-canonical Wnt signaling alters support from adjacent perivascular cells, which affects the response of endothelial cells to reduced flow.*

Although the comments on pericyte may appear relevant given the long standing idea that pericyte drop out precedes vessel regression in diabetic retinopathy, recent careful studies, including ours (Franco et al. PLoS Biol 2015) illustrate that pericytes generally cover regressing vessels, and even are left behind when the endothelial cells have retracted. Nevertheless, we have now quantified pericyte coverage of retinal blood vessels from Evi WT and Evi iEC-KO. We observed no difference in terms of presence or distribution of pericytes in the KO mouse line. Additionally, it is important to note that the increased shear sensitivity of the endothelial cells in the absence of Wnt is present in the in vitro assay, which lacks pericytes (new Figure 2). We are therefore confident the observed effects are endothelial cell-autonomous.

*Along the same lines, angiotensin II might affect pericyte migration and it is not clear that its effect can be attributed exclusively to changes in blood flow (have the authors actually measured blood flows upon addition of angiotensin?)*

In accordance to the previous reply, we did not observe any significant difference in pericyte distribution or numbers in Evi iEC-KO retinas. Pericyte migration is a very difficult aspect to analyze directly with currently available methods. The fact that pericytes are present all the way to the sprouting front in the mutant retinas, like in controls, strongly indicates however that there are no major pericyte migration defects. Animal models for direct dynamic visualization of pericytes in vivo are now becoming available, but retinal live imaging in development is not yet possible, and we lack the models to do this.

We tried to measure the effects of angiotensin II on blood flow parameters using the cranial window chamber assay. However, our current methodology did not allow us to obtain consistent and satisfactory measurements. We are currently investing into development of photoacoustic imaging to overcome this limitation, but until we get there, we will likely need another year. Thus, we cannot present quantitative information on this topic. Nevertheless, there are several papers presenting strong evidence that angiotensin II increases blood pressure (Timmermans et al. Pharmacol. Rev 1993; Gembardt et al. Faseb 2008).

*2) Specificity of the cre lines used: the authors should provide evidence to support the fact the knockouts are confined to ECs*

The specificity of the cre line used has been demonstrated many times in several previous reports, including the original paper describing this line (Claxton et al. Genesis 2008) and our previous report (Franco et al. Development 2013). Figure S1 from our previous report highlights the specificity of the Pdgfb-iCre line (http://dev.biologists.org/highwire/filestream/1203983/field_highwire_adjunct_files/0/DEV091074.pdf). Pericytes never show recombination with the Pdgfb-iCre line.

*3) Could the authors better link the two systems they use and somehow connect the observations* in vivo *and in the HUVEC cells? Do the changes in transcription in HUVECs actually match to any degree what is seen in vivo?*

Endothelial cells in vivo are rarely in static conditions. Therefore, we can only measure gene transcription levels of endothelial cells exposed to a mixture of shear stress levels. In these conditions, we observed no significant differences of gene expression of flow-regulated genes PTGS2, KLF2, and KLF4 in EVI iEC-KO compared to EVI WT mouse retinas (new Figure 5).

*Alternatively, could more physiological conditions be applied to the HUVEC assay? 20 dyn/cm2 is much higher than what is typically observed in vivo, would the changes in gene expression observed also occur for lower shear stresses?*

20dynes/cm2 are not uncommon values for WSS in vivo (Chatzizisis et al. J Am Coll Cardiol 2007; Gimbrone et al. Ann N Y Acad Sci 2000; Malek et al. JAMA 1999), and they are within the range of shear stress levels commonly used in the literature. Several microarray datasets (GSE8852; GSE13712; GSE16706; GSE29376) previously published showed that levels of expression of KFL2, KLF4 and PTGS2 are significantly increased upon low or high shear stress conditions.

*Could the authors also provide some images of the HUVEC system so the reader gets a better idea of what is meant by polarity in Figure 7?*

We have now included in new Figure 8 images of HUVECs for the various conditions.

*4) In Figure 9 the authors report that overexpressing Wnt5a does not prevent vessel regression. How about in the presence of angiotensin? Would overexpression of Wnt5a rescue the effect of increased blood flow?*

We have now quantified vessel regression in Wnt5a GOF following Angiotensin treatment. We included the quantification in the new Figure 10. We observed no significant differences in terms of vessel regression compared to control conditions. Thus, overexpression of Wnt5a does not prevent flow-dependent vessel regression.